# Parvovirus B19 Infection Is Associated with the Formation of Neutrophil Extracellular Traps and Thrombosis: A Possible Linkage of the VP1 Unique Region

**DOI:** 10.3390/ijms25189917

**Published:** 2024-09-13

**Authors:** Bor-Show Tzang, Hao-Yang Chin, Chih-Chen Tzang, Pei-Hua Chuang, Der-Yuan Chen, Tsai-Ching Hsu

**Affiliations:** 1Institute of Medicine, Chung Shan Medical University, Taichung 402, Taiwan; bstzang@csmu.edu.tw (B.-S.T.); j104023@gmail.com (H.-Y.C.); dylan.ymu@gmail.com (P.-H.C.); 2Department of Biochemistry, School of Medicine, Chung Shan Medical University, Taichung 402, Taiwan; 3Department of Clinical Laboratory, Chung Shan Medical University Hospital, Taichung 402, Taiwan; 4Immunology Research Center, Chung Shan Medical University, Taichung 402, Taiwan; 5School of Medicine, College of Medicine, National Taiwan University, Taipei City 100, Taiwan; jerrytzang@gmail.com; 6College of Medicine, China Medical University, Taichung 404, Taiwan; 7Rheumatology and Immunology Center, China Medical University Hospital, Taichung 404, Taiwan

**Keywords:** human parvovirus B19 (B19V), neutrophil extracellular traps (NETs), VP1 unique region (VP1u) IgG, antiphospholipid antibodies (aPLs), thrombosis

## Abstract

Neutrophil extracellular traps (NETs) formation, namely NETosis, is implicated in antiphospholipid syndrome (APS)-related thrombosis in various autoimmune disorders such as systemic lupus erythematosus (SLE) and APS. Human parvovirus B19 (B19V) infection is closely associated with SLE and APS and causes various clinical manifestations such as blood disorders, joint pain, fever, pregnancy complications, and thrombosis. Additionally, B19V may trigger the production of autoantibodies, including those against nuclear and phospholipid components. Thus, exploring the connection between B19V, NETosis, and thrombosis is highly relevant. An in vitro NETosis model using differentiated HL-60 neutrophil-like cells (dHL-60) was employed to investigate the effect of B19V-VP1u IgG on NETs formation. A venous stenosis mouse model was used to test how B19V-VP1u IgG-mediated NETs affect thrombosis in vivo. The NETosis was observed in the dHL-60 cells treated with rabbit anti-B19V-VP1u IgG and was inhibited in the presence of either 8-Br-cAMP or CGS216800 but not GSK484. Significantly elevated reactive oxygen species (ROS), myeloperoxidase (MPO), and citrullinated histone (Cit-H3) levels were detected in the dHL60 treated with phorbol myristate acetate (PMA), human aPLs IgG and rabbit anti-B19V-VP1u IgG, respectively. Accordingly, a significantly larger thrombus was observed in a venous stenosis-induced thrombosis mouse model treated with PMA, human aPLs IgG, rabbit anti-B19V-VP1u IgG, and human anti-B19V-VP1u IgG, respectively, along with significantly increased amounts of Cit-H3-, MPO- and CRAMP-positive infiltrated neutrophils in the thrombin sections. This research highlights that anti-B19V-VP1u antibodies may enhance the formation of NETosis and thrombosis and implies that managing and treating B19V infection could lower the risk of thrombosis.

## 1. Introduction

Antiphospholipid syndrome (APS) is known as a systemic autoimmune disorder characterized by the presence of various autoantibodies that lead to a variety of complications such as pregnancy issues, stroke, deep vein thrombosis (DVT), and myocardial infarction (MI) [1]. Indeed, evidence has indicated that APS affects approximately two in one thousand individuals, making it a primary cause of pregnancy loss and thrombosis, especially in patients under 50 years of age [2,3]. APS commonly occurs alongside other autoimmune disorders such as systemic lupus erythematosus (SLE), anti-neutrophil cytoplasmic antibody (ANCA) related vasculitis, and rheumatoid arthritis (RA) [4,5,6,7]. Diagnosis of APS in clinical settings typically involves the detection of antiphospholipid antibodies (aPLs), such as anti-cardiolipin antibodies (aCL), anti-β2GPI antibodies, and lupus anticoagulants [8].

Research has established that neutrophil extracellular traps (NETs) play essential roles in developing thrombosis in APS [4,7]. Experiments conducted with APS or thrombosis animal models have demonstrated that administering deoxyribonuclease or inhibitors to prevent NET release reduces thrombosis formation and inhibits neutrophil cell death (NETosis) [9]. Additionally, the presence of APS, NET formation, and thrombosis has also been observed in human patients, with NETs formation found within thrombi [10] Notably, anti-β2GPI antibody, a common type of anti-phospholipid antibody (aPLs), is found to induce NETosis directly [9,11,12]. The plasma of APS patients and isolated aPLs can stimulate NET release from healthy neutrophils, with the release of NETs correlating with aPLs activity [10,13,14,15]. Subsequently, the immune system can also recognize the components of NETs, causing a pro-coagulative cascade, ultimately leading to thrombus formation [10,12]. The process of NETosis involves both PAD4-dependent and PAD4 (Peptidylarginine Deiminase 4)-independent mechanisms [16]. PAD4 is an enzyme that catalyzes the conversion of arginine residues in proteins to citrulline, a process known as citrullination. Citrullination of histones by PAD4 is essential for chromatin decondensation, a crucial step in NET formation during suicidal NETosis. Inhibition of PAD4 can prevent NET formation, making it a potential therapeutic target for diseases characterized by excessive NETosis [17]. Alternatively, the cAMP/PKA (cyclic AMP/Protein Kinase A) signaling pathway regulates NETosis through the modulation of reactive oxygen species (ROS) production via the NADPH oxidase complex, offering an independent route for NET formation that underscores the complexity and redundancy of NETosis mechanisms [16]. While NETosis is a critical immune defense mechanism, its excessive or dysregulated activity is implicated in various diseases, including autoimmune disorders, thrombosis, and sepsis [4,5,6,7]. Therefore, understanding the pathways involved in NETosis is crucial for developing targeted therapies to modulate this process and treat related diseases.

Human parvovirus B19 (B19V), a non-enveloped single-stranded DNA virus, belongs to the Parvoviridae family, mainly composed of two structural proteins, VP1, VP2, and non-structural protein NS1 [18]. A 227-residue region located in the N-terminal of B19V-VP1 is known as the B19V-VP1 unique region (VP1u). The sequence of B19V-VP1u contains a phospholipase A2 motif, which exhibits secretory phospholipase A2 (sPLA2) activity and is essential for the internalization of B19V and the successful host infection [19]. Previous studies have indicated that B19V infection is closely related to various autoimmune diseases, of which SLE, RA, APS, and vasculitis are the most common [20]. A previous study has shown that B19V infection can trigger the production of aPLs [21]. Various studies have also indicated that antibodies against B19V-VP1u exhibit significant cross-reactivity with β2GPI epitopes, likely attributable to the external positioning and phospholipase region of B19V-VP1u [22,23,24]. Notably, B19V-VP1u antibodies and aPLs have revealed similar pathogenic activities, such as inducing inflammation in endothelial cells and leading to a pro-coagulative state [25,26,27].

Although the B19V-VP1u antibodies exhibit comparable thrombotic effects to aPLs [23,24,25,28], the precise role and underlying mechanism of B19V-VP1u antibodies in developing thrombosis remains unclear. For further exploring the influence of B19V-VP1u antibodies in thrombus formation, this study employs a narrow thrombus mouse model [29] to determine whether B19V infection-induced thrombosis involves the activation of a NETosis pathway. To address how B19V-VP1u antibodies contribute to NETosis, we applied different inhibitors targeting various stages of NETs production. This approach was designed to dissect the specific mechanisms involved and to clarify whether the observed thrombotic effects are indeed mediated through NETosis activation.

## 2. Results

### 2.1. B19V-VP1u IgG Stimulates NETs Formation in dHL-60 Neutrophils

To verify the influence of IgG against human parvovirus B19V-VP1u on the formation of neutrophil extracellular traps (NETs), differentiated HL60 neutrophils (dHL-60) were obtained with DMSO stimulation and subsequently treated with rabbit anti-B19V-VP1u IgG. The generation of NETs was observed in dHL-60 treated with 25 nM PMA, a well-characterized NETosis inducer, and 100 μg rabbit anti-B19V-VP1u IgG by SYTOX Green staining. In contrast, no NETs release was observed in dHL-60 cells treated with 100 μg rabbit IgG (Figure 1). For further confirming the role of rabbit anti-B19V-VP1u IgG on formation of NETs, various NETosis inhibitors, including 8-Br-cAMP, CGS21680 and GSK484, were adopted to treat dHL-60 with either 25 nM PMA or 100 μg rabbit anti-B19V-VP1u IgG. In the presence of 8-Br-cAMP, CGS21680 or GSK484, no NETs formation was observed in dHL-60 treated with PMA (Figure 2A). Notably, generation of NETs was observed in dHL-60 treated with 100 μg rabbit VP1u IgG in the presence of GSK484. In contrast, no NETs formation was detected in the presence of either 8-Br-cAMP or CGS21680 (Figure 2B).

### 2.2. B19V-VP1u IgG Stimulates the Expressions of Cit-H3 and MPO in dHL-60 Neutrophils

To verify the influence of rabbit anti-B19V-VP1u IgG on expressions of NETosis markers, quantitative cytofluorimetric analysis by binding free citrullinated histone (Cit-H3) and myeloperoxidase (MPO) was performed. Significantly higher expressions of both Cit-H3 and MPO were detected in dHL-60 neutrophils treated with 25 nM PMA, 100 μg human aPLs IgG, and 100 μg rabbit anti-B19V-VP1u IgG as compared to the control, human IgG, and rabbit IgG, respectively (Figure 3). Immunoblotting was adopted to confirm the expression of Cit-H3 further. Consistently, a significantly higher amount of Cit-H3 protein was detected in the dHL-60 neutrophils in the presence of PMA, human aPLs IgG and rabbit anti-B19V-VP1u IgG (rabbit VP1u IgG) as compared to the control, normal human IgG (NH IgG) and rabbit IgG, respectively (Figure 4). The reactive oxygen species (ROS) level was also measured to identify the role of rabbit anti-B19V-VP1u IgG on NETs generation. As the arrow indicated, significantly higher ROS levels were detected in the dHL-60 treated with PMA, human aPLs IgG, and rabbit VP1u IgG compared to the control, NH IgG, and rabbit IgG, respectively *(*Figure 5).

### 2.3. B19V-VP1u IgG Increases Thrombus Formation in Mice

A venous stenosis-induced thrombosis mouse model was adopted to verify the influence of B19V-VP1u IgG-mediated NETs on thrombosis in vivo. Compared with the mice receiving PBS or rabbit IgG, the mice receiving rabbit anti-B19V-VP1u IgG formed significantly larger thrombi in both weight and length (Figure 6A,B). Given that Cit-H3, MPO, and CRAMP are recognized as the most well-known biochemical markers of NETs, immunohistochemistry staining (IHC) for Cit-H3, MPO, and CRAMP was performed. A significantly higher amount of Cit-H3-, MPO- and CRAMP-positive infiltrated neutrophils were detected in the thrombi sections from the mice receiving rabbit anti-B19V-VP1u IgG as compared to those mice receiving PBS or rabbit IgG (Figure 6C–E). Additionally, a significantly larger weight and length thrombi were detected in the mice receiving human aPLs IgG and human anti-B19V-VP1u IgG compared to those receiving normal human IgG (Figure 7A,B). Accordingly, a significantly higher amount of Cit-H3-, MPO- and CRAMP-positive infiltrated neutrophils were detected in the thrombi sections from the mice receiving human aPLs IgG or human anti-B19V-VP1u IgG as compared with those receiving normal human IgG (Figure 7C–E).

## 3. Discussion

Although ongoing debate exists about whether the B19V can lead to thrombosis [30], various clinical cases have highlighted a correlation between B19V infection and thrombosis [31,32,33]. Thrombosis has been reported in both healthy and diseased individuals infected with B19V. A previous study indicated that a 25-year-old woman in good health experienced a middle cerebral artery thrombosis and subsequent stroke during a recurrence of B19V infection [31]. A similar result was observed in a healthy 47-year-old man with B19V infection, which was reported to reveal microthrombi in kidney vasculature, suggesting a thrombotic microangiopathy [32]. Notably, the sudden death of a 20-year-old Kawasaki disease (KD) patient with B19V infection by thrombotic occlusion of the coronary arteries was observed [33]. Interestingly, it is crucial to highlight that B19V infection frequently leads to thrombocytopenia [34]. Thrombocytopenia, characterized by low platelet levels in the blood, typically raises concerns about increased bleeding risk. However, certain types like immune thrombocytopenic purpura (ITP) and thrombotic thrombocytopenic purpura (TTP) paradoxically heighten the risk of thrombosis [35]. This seeming contradiction stems from intricate interactions within the blood coagulation system. In thrombocytopenia, the remaining platelets can become hyperactive or dysfunctional, potentially aiding clot formation and suggesting a compensatory mechanism that enhances clotting tendencies [36]. These findings provide a possible rationale for explaining the induction of thrombosis via B19V infection.

NETs play pivotal roles in abnormal thrombus formation in many disorders, which seriously threaten human health [11,37,38,39]. The causes of NETosis and thrombosis are diverse, and the most common ones include autoantibodies and microbial infections [39]. Generating reactive oxygen species (ROS) through NADPH oxidase is important in causing NETosis [40]. Additionally, the formation of citrullinated histone H3 (Cit-H3) by peptidyl arginine deiminase 4 (PAD4), which converts arginine residues to citrulline of histone proteins, causes significant chromatin decondensation and consequent nuclear delobulation and swelling during NETosis [41,42]. These findings indicated ROS and Cit-H3 as target biomarkers reflecting NET formation and thrombosis [43,44,45]. Notably, evidence has demonstrated that antiphospholipid antibodies (aPLs) and anti-β2GPI antibodies isolated from patients with antiphospholipid syndrome (APS) can induce human neutrophils to release NETs as well as massive expressions of ROS, Cit-H3, and myeloperoxidase (MPO) [10,13,14,15]. Interestingly, the current study, for the first time, demonstrated that both rabbit and human anti-B19V-VP1u IgG revealed similar activity as human aPLs by increasing NETs, Cit-H3, and MPO expressions in dHL-60 cells and inducing thrombus in mice. Although the reasons why anti-B19V-VP1u IgG exhibits characteristics similar to human aPLs and how it induces thrombosis are still not fully understood, certain earlier research may provide insights into this phenomenon. Previously, B19V infection has been associated with producing aPLs-like antibodies [46]. Autoantibodies against cardiolipin (CL), β2GPI, and phospholipid (PhL) isolated from patients with B19V infection were also found to reveal cross-reactivity with B19V-VP1u protein [24]. The mice immunized with anti-B19V-VP1u IgG developed thrombocytopenia, prolongation of aPTT, and autoantibodies against β2GPI and PhL [24,28]. Notably, ant-B19V-VP1u IgG showed a noteworthy cross-reactivity with β2GPI epitopes, possibly due to the external positioning and phospholipase region of B19V-VP1u [22,23,24]. These findings provide a compelling explanation for the ability of anti-B19V-VP1u IgG in NETs-mediated thrombosis, as these antibodies exhibit numerous similarities with human aPLs.

NETosis, namely the formation of NETs, is a significant defensive response of the immune system against various pathogens. The molecular pathways underlying NETs formation are multifaceted, contributing to antimicrobial actions and participation in non-infectious, autoimmune, and inflammatory conditions [47,48,49]. So far, several well-established pathways have been identified as being involved in NETosis, including NADPH oxidase (NOX)-dependent pathway, peptidylarginine deiminase 4 (PAD4) pathway, mitogen-activated protein kinases (MAPKs) pathway, intracellular calcium signaling, autophagy machinery pathway, Toll-like receptors (TLRs) signaling, and Syk (Spleen Tyrosine Kinase)/PI3K (Phosphoinositide 3-Kinase) pathway [47,48].

To verify the involvement of NETosis, it is essential to inhibit NETosis using blockers targeting the molecules in these signaling pathways. Three commonly used inhibitors for NETosis, including 8-Br-cAMP (cAMP inhibitor), CGS21680 (selective agonist of the adenosine A2A receptor) and GSK484 (PAD4 inhibitor) [9], were used to confirm the NETosis induced by rabbit anti-B19V-VP1u IgG. PMA was utilized as a positive control since evidence has shown that PMA can trigger NETosis by elevating ROS and MPO levels, and the PMA-induced NETosis can be suppressed by 8-Br-cAMP, CGS21680, and GSK484 [9]. This study observed significantly increased ROS, MPO, and NETosis levels in dHL-60 cells treated with rabbit anti-B19V-VP1u IgG. However, rabbit anti-B19V-VP1u IgG-induced NETosis in dHL-60 cells disappeared in either 8-Br-cAMP or CGS21680 but not in the presence of GSK484. Since ROS production can trigger the release of MPO and neutrophil elastase (NE), resulting in the subsequent initiation of PAD4-independent NETosis [49,50], it suggests that anti-B19V-VP1u IgG-induced NETosis may involve a PAD4-independent pathway. The results suggest that the cAMP/PKA pathway, rather than PAD4-mediated NETosis, may play a more central role in suppressing thrombosis in the context of B19V-VP1u IgG-induced NETosis. This is consistent with research indicating that PAD4-independent mechanisms are significant in NET formation and relevant in autoimmune conditions like lupus [51] and rheumatoid arthritis, ref. [52] both of which are associated with B19V [53]. Accordingly, a recent study comparing aPLs in patients with refractory obstetric APS and patients with other clinical manifestations of APS showed that aPLs antibodies isolated from different groups of APS patients differentially contribute to endothelial activation and dysfunction [54]. Therefore, our findings imply that B19V-VP1u-associated thrombosis may involve mechanisms beyond PAD4-mediated NETosis, potentially through ROS production and other processes regulated by the cAMP/PKA pathway. There is evidence to suggest that these effects, such as the selective agonism of the adenosine A2A receptor by CGS21680 and the action of 8-Br-cAMP through cAMP pathways, could also be beneficial in preventing thrombosis in vivo, as both have been shown to reduce aPL antibody-mediated NETosis and thrombosis in murine experimental models [9].

Evidence has indicated that the generation of thrombosis in APS may arise from a combination of factors such as activated endothelial cells, myeloid-lineage cells, complement cascade, coagulation, and fibrinolytic systems, along with the formation of neutrophil extracellular traps (NETs) [55]. Therefore, an appropriate animal model is crucial in studying APS’s thrombus generation. For decades, animal models of thrombosis have played a critical role in advancing understanding within the APS field, which has prompted reevaluations in patient care [56,57]. Regarding evaluating venous thrombosis, these traditional animal models have depended on causing direct damage to the vessel wall. However, it is not a common aspect of the natural development of deep vein thrombosis in humans, with blood flow stasis being the more important contributing factor [58]. Therefore, the current study adopted a mice model of venous thrombosis achieved by decreasing blood flow through the inferior vena cava (IVC) [10]. The stenosis thrombosis model is crucial because it provides a stable core for thrombus development in the IVC, allowing the thrombus to remain in a fixed position. This stability is advantageous for subsequent biopsy analysis and quantification, as it ensures that the thrombus structure remains intact. Moreover, this model is a better simulation of human thrombosis and reduces possible interference caused by injury to vascular endothelial cells, making it highly relevant for translational research [10]. It enables researchers to observe and investigate the early cellular and molecular events during the initiation of venous thrombosis [29]. On the contrary, direct injection of antibodies without an existing thrombosis model may lead to disseminated thrombi, complicating observation. As autoantibody-induced thrombosis is typically a chronic autoimmune process, this approach may not be effective for studying acute inflammatory and coagulative events.

Recent guidelines for in vivo ROS measurement stress the importance of carefully evaluating the chemistry, selectivity, and potential artifacts when using fluorescent probes like DCFH-DA. Proper controls are essential to confirm that the observed effects are due to the targeted ROS species, and orthogonal techniques should be used for validation [59]. In this study, controls were applied to comparisons such as NH IgG vs. human aPLs IgG and Rabbit IgG for VP1u IgG. Orthogonal methods further supported biological evidence, including increased MPO activity and inhibition of thrombosis by CGS21680, an MPO pathway mediator. Future research should incorporate ROS blocker tests and ROS-specific markers to directly validate the hypothesis that MPO generates HOCl and H2O2. Additionally, it is also beneficial to directly quantify NETs using the MPO-DNA complex by ELISA, a well-established technique that could further validate our findings on B19V-VP1u IgG-induced NETosis. Moreover, while the HL60 cell line provides significant advantages in terms of reproducibility, accessibility, and extended lifespan, it is recognized that using this cell line has limitations [60]. To improve the generalizability of our findings and better adapt them to human scenarios, future experiments should involve primary neutrophils from human volunteers or mice. Additionally, using NETosis inhibitors such as 8-Br-cAMP, CGS21680, and GSK484 to verify the effect of B19V-VP1u on thrombosis in animal experiments should be considered. These would help confirm the relevance of our results and provide a more accurate representation of the biological processes in vivo.

## 4. Materials and Methods

### 4.1. Cell Culture

Human promyeloblast HL60 cells were purchased from the Bioresource Collection and Research Center (BCRC) at the Food Industry Research and Development Institute (FIRDI), Hsinchu, Taiwan. The cells were cultured in RPMI 1640 medium supplemented with 10% (*v*/*v*) heat-inactivated calf serum (FCS) at 37 °C and 5% CO_2_. Differentiation of HL-60 cells into neutrophil-like cells (dHL-60) was conducted by treatment with 1.3% DMSO in RPMI 1640 for 7 days [61]. The morphological alterations of HL60 cells following DMSO treatment were examined using Giemsa stain.

### 4.2. Purification of Immunoglobulin G (IgG)

Rabbit anti-B19V-VP1u IgG (Rabbit VP1u IgG) was purified as described in our previous study [62]. Briefly, protein G agarose chromatography (Roche Applied Science, Indianapolis, IN, USA) was used to purify the rabbit VP1u IgG and the rabbit control IgG. The refined antibodies were filtered through an EMD Millipore 0.22 μm microporous membrane. A U3000 spectrophotometer (Hitachi, Ltd., Tokyo, Japan) was used for quantification. Normal human serum IgG (NH IgG; 100 µg/mL) and human aPLs IgG (100 µg/mL; 250 units) were obtained from a commercial APhL ELISA IgG HRP kit (cat. nos. LAPLK-HRP-001G; Louisville APL Diagnostics, Inc., Louisville, KY, USA).

### 4.3. Immunofluorescence Staining for NETosis

Fluorescence microscopy was employed to assess the formation of neutrophil extracellular traps (NETs) as described elsewhere [10,63]. In brief, 5 × 10^5^ dHL-60 cells were seeded onto 24-well tissue culture plates coated with 1% poly-L-lysine and incubated at 37 °C with 5% CO_2_. Subsequently, the cells were treated with 25 nM phorbol myristate acetate (PMA), 100 μg/mL NH IgG/aPLs IgG/rabbit IgG/rabbit VP1u IgG for 7 h. Following treatment, the cells were stained with Sytox Green cell-impermeable nucleic acid dye (1 μM, Thermo Fisher Scientific, Waltham, MA, USA) and nuclear stain Hoechst 33342. The stained cells were then examined using fluorescence microscopy. For NETosis inhibition assays, inhibitors were added 1 h before stimulation at the following final concentrations: 10 μM 8-Br-cAMP (cAMP inhibitor), 10 nM CGS21680 (selective agonist of the adenosine A2A receptor), and 10 μM GSK484 (PAD4 inhibitor) (Sigma-Aldrich, St. Louis, MO, USA). The lowest effective concentrations of NETosis inhibitors were determined based on a previous report [9].

### 4.4. Flow Cytometry

A flow cytometric assay was conducted to assess NET formation. Neutrophil-like dHL-60 cells (2 × 10^6^) were seeded onto a 24-well cell culture plate and incubated for 1 h in a CO_2_ incubator at 37 °C. The cells were treated with 25 nM PMA, 100 μg/mL NH IgG/human aPLs IgG/rabbit IgG/rabbit VP1u IgG for 7 h, respectively. Subsequently, the cells were fixed in 2% paraformaldehyde and then blocked for 30 min with 2% bovine serum albumin (BSA, Sigma-Aldrich, St. Louis, MO, USA) in Dulbecco’s phosphate-buffered saline (DPBS) at 37 °C. Before flow cytometry analysis, the cells were incubated with the primary citrullinated histone H3 antibody (Cit-H3, Abcam), Alexa Fluor 700-conjugated secondary antibody (Thermo Scientific), and FITC-conjugated anti-myeloperoxidase (anti-MPO) antibody (Abcam, Cambridge, UK) at 37 °C for 1 h in the dark.

### 4.5. Immunoblotting

Immunoblotting was performed using whole-cell lysates from dHL-60 cells. Briefly, 15 μg protein samples were separated in a 15% SDS-PAGE and electrophoretically transferred to an Immobilon-E PVDF membrane (Merk Millipore Ltd., Carrigtwohill, County Cork, Ireland). After blocking with 5% non-fat dry milk for 1 h at 4 °C, antibodies against Histone-3, citrullinated histone-H3 antibody (Cit-H3, Abcam), and β-actin (EMD Millipore) were incubated with the membrane for 3 h with gentle agitation at room temperature. Horseradish peroxidase (HRP)-conjugated secondary antibodies (Santa Cruz Biotechnology, Inc., Dallas, TX, USA) were added and incubated for another 1 h at 25 °C. Finally, an Immobilon Western Chemiluminescent HRP Substrate (EMD Millipore) and a chemiluminescence imaging analyzer (GE ImageQuant TL 8.1, GE Healthcare Life Sciences, PA, USA) were used to detect the antigen-antibody complexes.

### 4.6. Determination of Reactive Oxygen Species (ROS) Production

A total of 2 × 10^6^ cells were treated with either 25 ng/mL PMA or 100 μg/mL NH IgG/human aPLs IgG/rabbit IgG/rabbit VP1u IgG for 7 h. Subsequently, the cells were further incubated for 30 min at 37 °C in the presence of 5 μM DCFH-DA (Dichloro-dihydro-fluorescein diacetate) and then subjected to flow cytometry analysis.

### 4.7. Animal Model and Ethical Approval

Twenty-four male C57BL/6 mice at the age of 6 weeks were acquired from the National Laboratory Animal Center in Taiwan. The Institutional Animal Care and Use Committee (IACUC approval no. 2435 and 2612) of Chung Shan Medical University in Taiwan approved the study protocols. Each animal had free access to water and standard laboratory chow (Lab Diet 5001; PMI Nutrition International Inc., Brentwood, MO, USA) in a controlled lighting, temperature, and humidity room. At the age of ten weeks, the mice were randomly split into six groups: PBS (control), rabbit IgG, rabbit VP1u IgG, NH IgG, human aPLs IgG, and human B19V-VP1u IgG groups.

### 4.8. Human Serum Sample and Ethical Approval

Human serum samples were drawn from five patients diagnosed with autoimmune or autoinflammatory diseases, including two SLE patients, two RA patients, and one patient with adult-onset Still’s disease, between September 2021 and May 2023. The blood samples were tested at the Rheumatology and Immunology Center of the China Medical University Hospital, Taichung, Taiwan. Each study participant provided written informed consent. The Chung Shan Medical University Hospital’s Institutional Review Board approved the current study (IRB approval no. CS1-20208), and all participants provided written informed consent following the Declaration of Helsinki’s ethical guidelines for medical research involving human subjects. Samples of peripheral blood were drawn without using an anticoagulant or additive. Every subject in which B19V IgM/IgG ELISA kits (IBL, Hamburg, Germany), B19V-VP1u IgG antibodies by Western blot, and B19V DNA by nested PCR were used to detect B19V infection. The five positive serums of human B19V-VP1u IgG were purified by protein G agarose chromatography and filtered through a 0.22 µm microporous membrane (EMD Millipore) for thrombus assay.

### 4.9. In Vivo Venous Thrombosis

A narrowed IVC thrombosis mouse model was conducted to simulate the formation of human thrombosis as described elsewhere [10,11,27]. Male C57BL/6 wild-type mice were intraperitoneally injected twice with PBS, rabbit IgG, rabbit VP1u IgG, NH IgG, human aPLs IgG, and human B19V-VP1u IgG + (500 μg/100ul each time) 48 h apart. A laparotomy under anesthesia was performed before the second injection, and a blunted 30-gauge needle was wrapped around the IVC to create a ligature. Two days after the laparotomy, the mice were sacrificed with CO2 euthanasia, and the thrombus was harvested for further evaluation.

### 4.10. Thrombus Sectioning, HE Stain, and Immunohistochemistry

After the thrombus was fixed in 10% neutral buffered formalin and subsequently embedded in paraffin, the thrombus was then cut into 5 μm sections. Hematoxylin and eosin (HE) staining was performed by immersing tissue sections in Hematoxylin for 3 min, followed by rinsing in distilled water and sequential immersion in high concentrations of ethanol, then xylene for 30 s each, and finally air drying. For IHC staining, tissue sections were fixed in acetone, followed by air dried, and then treated with 0.3% H_2_O_2_/PBS to inhibit peroxidase activity. A rabbit anti-mouse myeloperoxidase (MPO) polyclonal antibody (1:500) (ab208670, Abcam, Cambridge, MA, USA) was utilized to identify neutrophils. Rabbit anti-mouse CRAMP polyclonal antibody (1:100) (A1640, Abclonal, Woburn, MA, USA) and rabbit polyclonal antibody against citrullinated histone H3 (Cit-H3) (1:100) (ab5103, Abcam, Cambridge, MA) were employed to detect NET-related markers [63,64].

### 4.11. Statistical Analysis

The statistical significance among experimental groups was evaluated using GraphPad Prism 5 software (Version 5.0, GraphPad Software, Inc., La Jolla, CA, USA) by one-way analysis of variance followed by Tukey’s multiple comparisons test. All data are presented as the mean ± standard error of the mean. Three independent experiments were performed. *p* < 0.05 was considered to indicate a statistically significant difference.

## 5. Conclusions

Thrombosis has significant impacts and influences in clinical settings, such as deep vein thrombosis (DVT), chronic thromboembolic pulmonary hypertension (CTEPH), pulmonary embolism (PE), and increased stroke risk and mortality, indicating far-reaching impacts in clinical practice, affecting patient outcomes, healthcare resource utilization in a variety of diseases [65,66]. Although currently there is no evidence directly proving that B19V-VP1u antibodies can induce thrombosis, this study highlights that antibodies against B19V-VP1u could enhance NETosis and subsequent thrombosis by elevating the levels of ROS, Cit-H3, and MPO. Effectively controlling B19V infection could potentially reduce the risk of thrombosis, thereby decreasing the need for medical interventions and lessening the related societal and healthcare burdens.

## Figures and Tables

**Figure 1 ijms-25-09917-f001:**
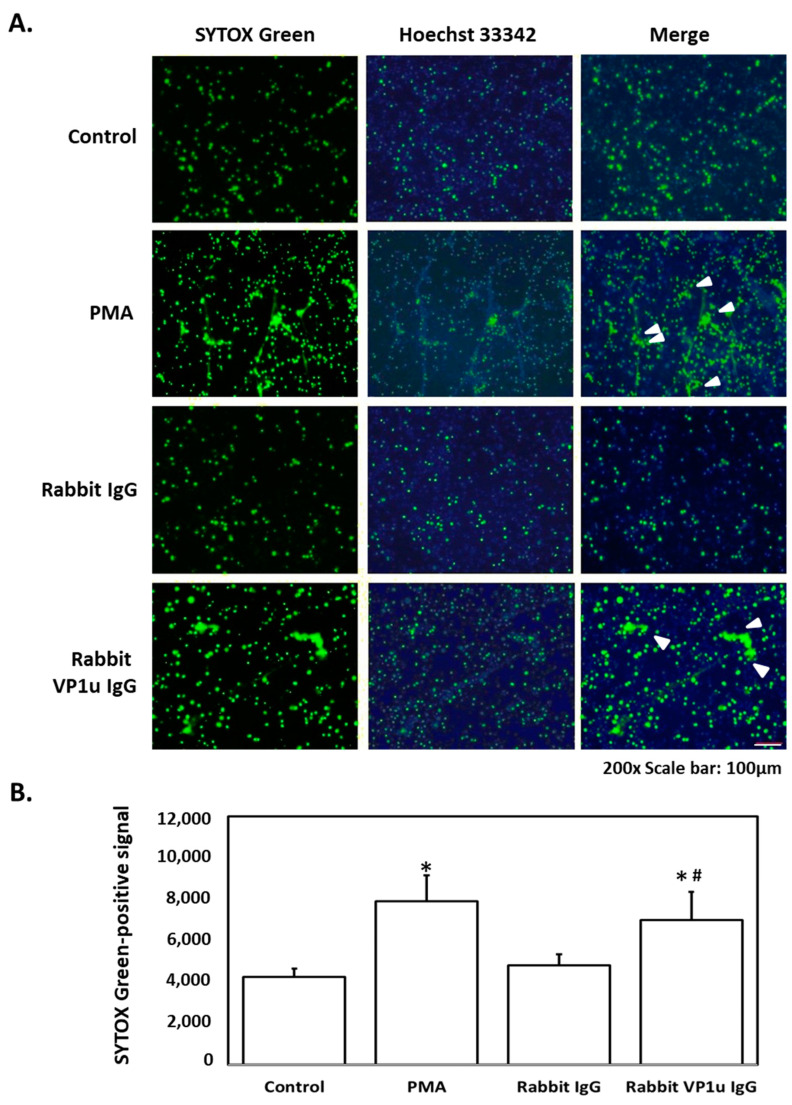
Rabbit anti-B19 VP1u IgG induces NETs release. The dHL-60 cells treated with PMA, rabbit IgG, and rabbit anti-B19V-VP1u IgG (rabbit VP1u IgG), respectively, were stained with (**A**) SYTOX Green (left panel) and Hoechst 33342 (middle panel). The merged images were shown in the right panel, and arrows indicated the NETs. (**B**) The quantified results of SYTOX Green-positive signal. Three independent experiments were performed. The symbols * and # indicate significance (*p* < 0.05) as compared to the control and rabbit IgG, respectively. *p*-value is calculated by one-way ANOVA followed by Tukey’s multiple comparison test.

**Figure 2 ijms-25-09917-f002:**
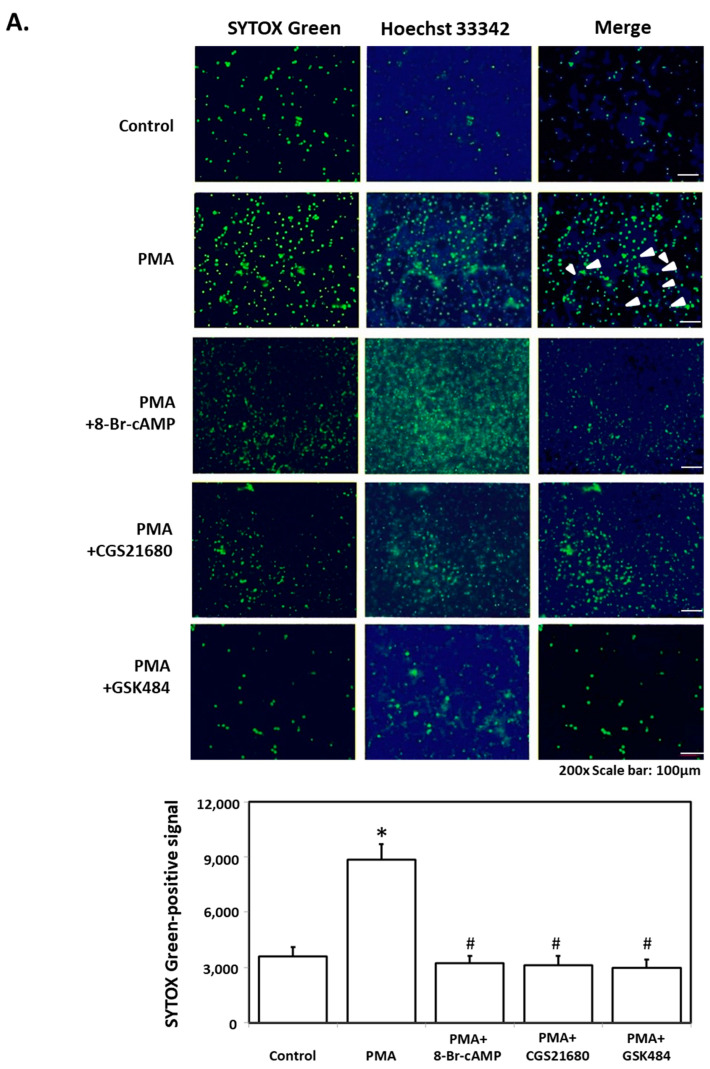
Inhibitory effect of NETs inhibitors (8-Br-cAMP, CGS21680, GSK484) on PMA and rabbit anti-B19V-VP1u IgG induced NETs. (**A**) The PMA and (**B**) rabbit anti-B19V-VP1u IgG (rabbit VP1u IgG) treated-dHL-60 cells in the presence of 8-Br-cAMP, CGS21680 or GSK484 were stained with SYTOX Green (left panel) and Hoechst 33342 (middle panel). The merged images were shown in the right panel, and arrows indicated the NETs. The quantified results of SYTOX Green-positive signals were shown in the lower panel. Three independent experiments were performed. The symbols *, #, and $ indicate significance (*p* < 0.05) as compared to the control, PMA, and rabbit anti-B19V-VP1u IgG, respectively. *p*-value is calculated by one-way ANOVA followed by Tukey’s multiple comparison test.

**Figure 3 ijms-25-09917-f003:**
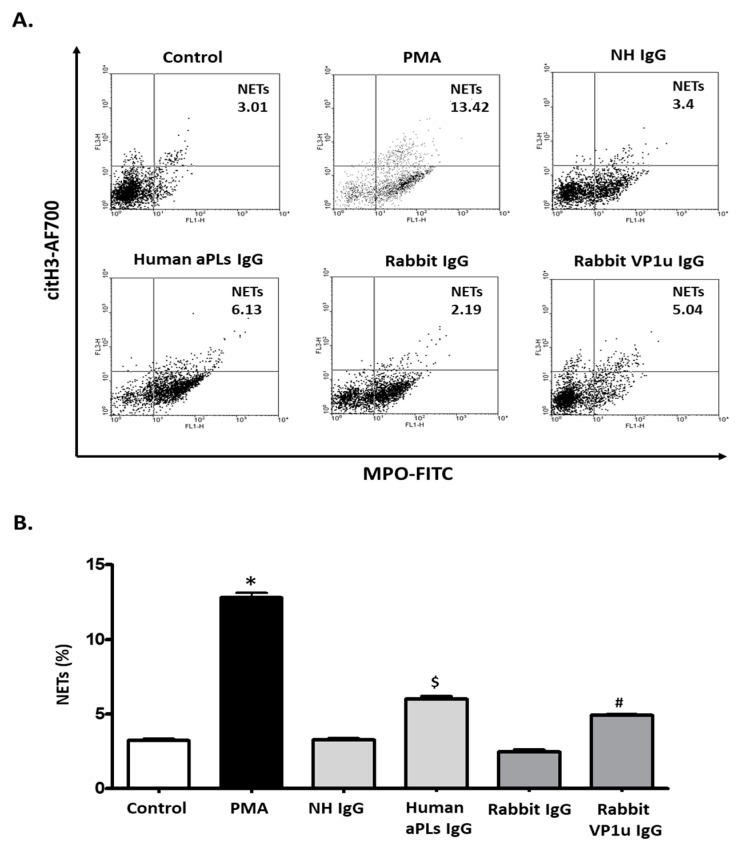
Rabbit B19V-VP1u IgG increases citH3 and MPO expressions. (**A**) The representative results of the dHL-60 cells were treated with PMA, normal human IgG (NH IgG), human aPLs IgG, rabbit IgG, and rabbit anti-B19V-VP1u IgG (rabbit VP1u IgG), and the presence of NETs was measured by detecting the expressions of citH3 and MPO with flow cytometry (**B**) The quantified results of NETs. Three independent experiments were performed. The symbols *, $, and # indicate significance (*p* < 0.05) as compared to the control, human IgG, and rabbit anti-B19V-VP1u IgG, respectively. *p*-value is calculated by one-way ANOVA followed by Tukey’s multiple comparison test.

**Figure 4 ijms-25-09917-f004:**
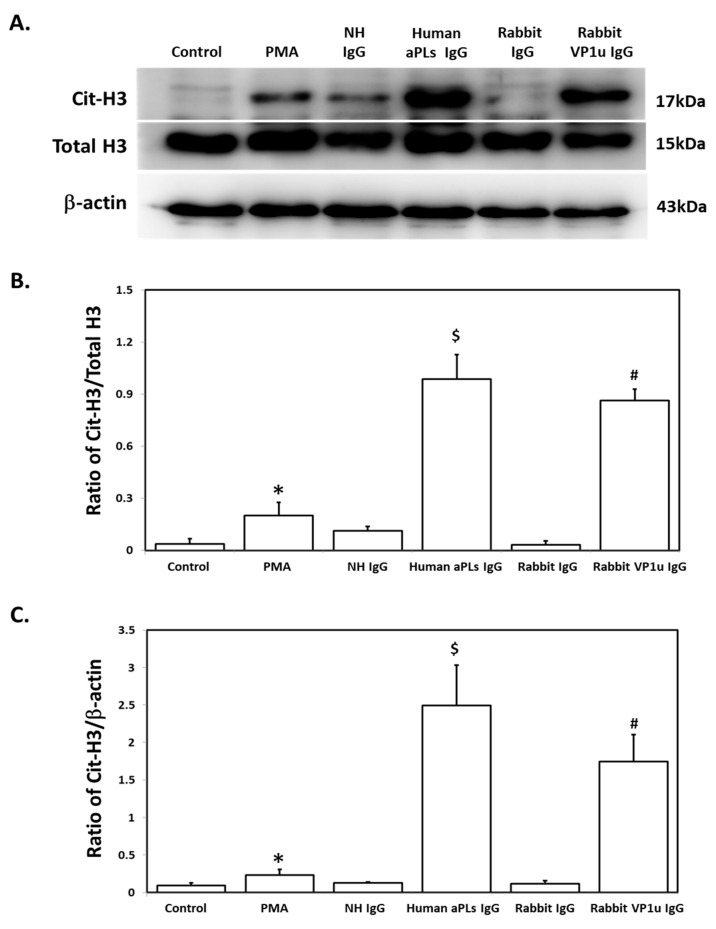
Rabbit anti-B19V-VP1u IgG increases citrullinated histone H3 (Cit-H3) expression. (**A**) Western blot analysis was used to detect the presence of Cit-H3 in dHL-60 cells treated with PMA, normal human IgG (NH IgG), human aPLs IgG, rabbit IgG, and rabbit anti-B19V-VP1u IgG (rabbit VP1u IgG). (**B**) The ratio of Cit-H3 amount relative to total H3. (**C**) The ratio of Cit-H3 amount relative to β-actin. Three independent experiments were performed. The symbols *, $, and # indicate significance (*p* < 0.05) compared to the Control, NH IgG, and rabbit VP1u IgG, respectively. *p*-value is calculated by one-way ANOVA followed by Tukey’s multiple comparison test.

**Figure 5 ijms-25-09917-f005:**
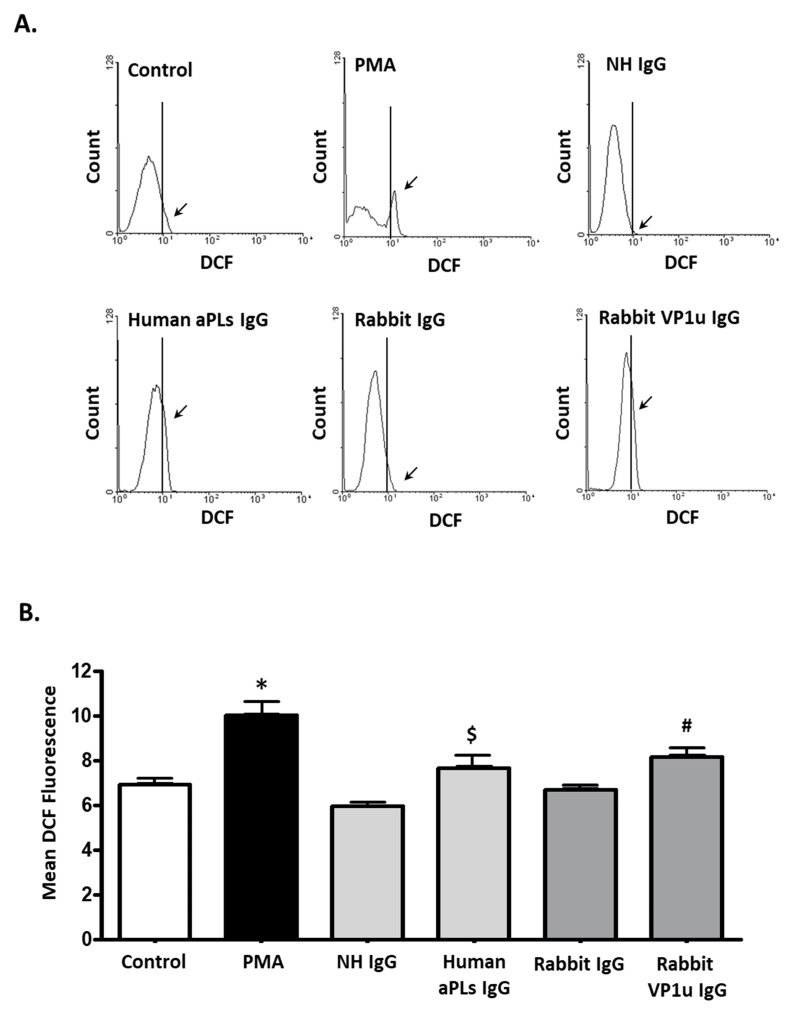
Human aPLs IgG and rabbit anti-B19V-VP1u IgG increase ROS production. (**A**) The dHL-60 cells were treated with PMA, normal human IgG (NH IgG), human aPLs IgG, rabbit IgG, and rabbit anti-B19V-VP1u IgG (rabbit VP1u IgG), and the ROS level was measured in the presence of Dichloro-dihydro-fluorescein diacetate (DCFH-DA) with flow cytometry. The arrow indicates the proportion of DCF-positive cells, defined as cells exhibiting fluorescence intensity greater than the established threshold value. (**B**) The quantified results of mean DCF. Three independent experiments were performed. The symbols *, $, and # indicate significance (*p* < 0.05) compared to the control, NH IgG, and rabbit VP1u IgG, respectively. *p*-value is calculated by one-way ANOVA followed by Tukey’s multiple comparison test.

**Figure 6 ijms-25-09917-f006:**
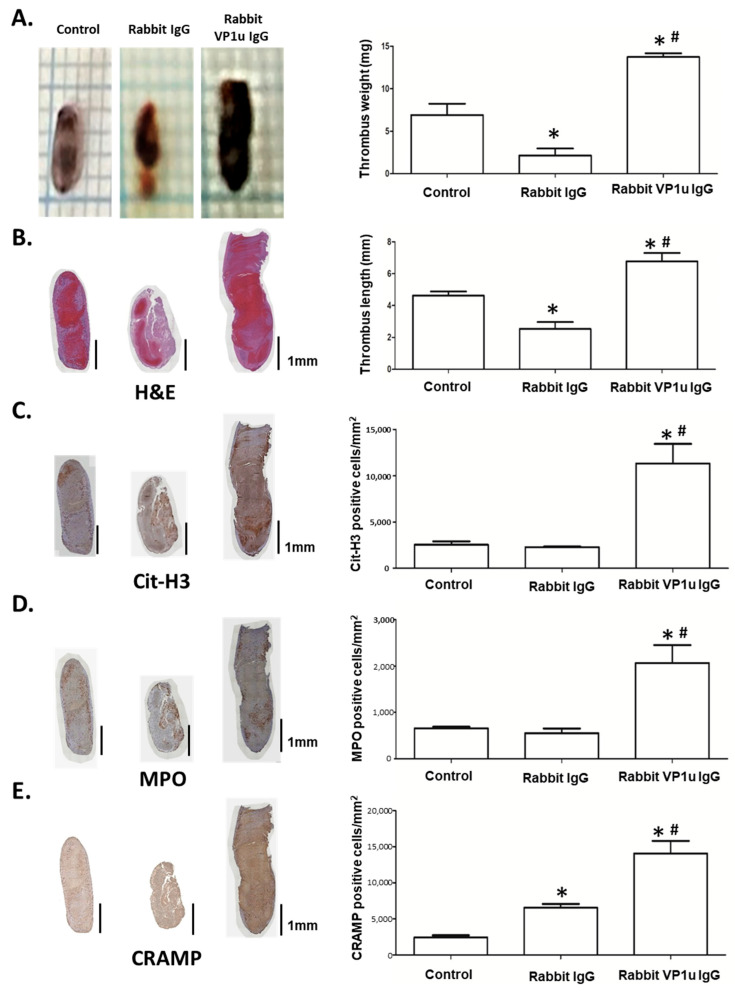
Rabbit anti-B19V-VP1u IgG promotes venous thrombosis in C57BL/6 mice with inferior vena cava ligation. (**A**) The representative images of thrombus from the mice treated with PBS, rabbit IgG, and rabbit anti-B19V-VP1u IgG (rabbit VP1u IgG). Sections of the thrombus stained with (**B**) H&E, (**C**) anti-citrullinated histone H3 (Cit-H3), (**D**) MPO, and (**E**) CRAMP. Scale bar = 1 mm. The right panel showed the quantified results of thrombus weight, thrombus length, and positive cells of Cit-H3, MPO, and CRAMP signal. The symbol * and # indicate significance (*p* < 0.05) as compared to the control and rabbit IgG, respectively. *p*-value is calculated by one-way ANOVA followed by Tukey’s multiple comparison test.

**Figure 7 ijms-25-09917-f007:**
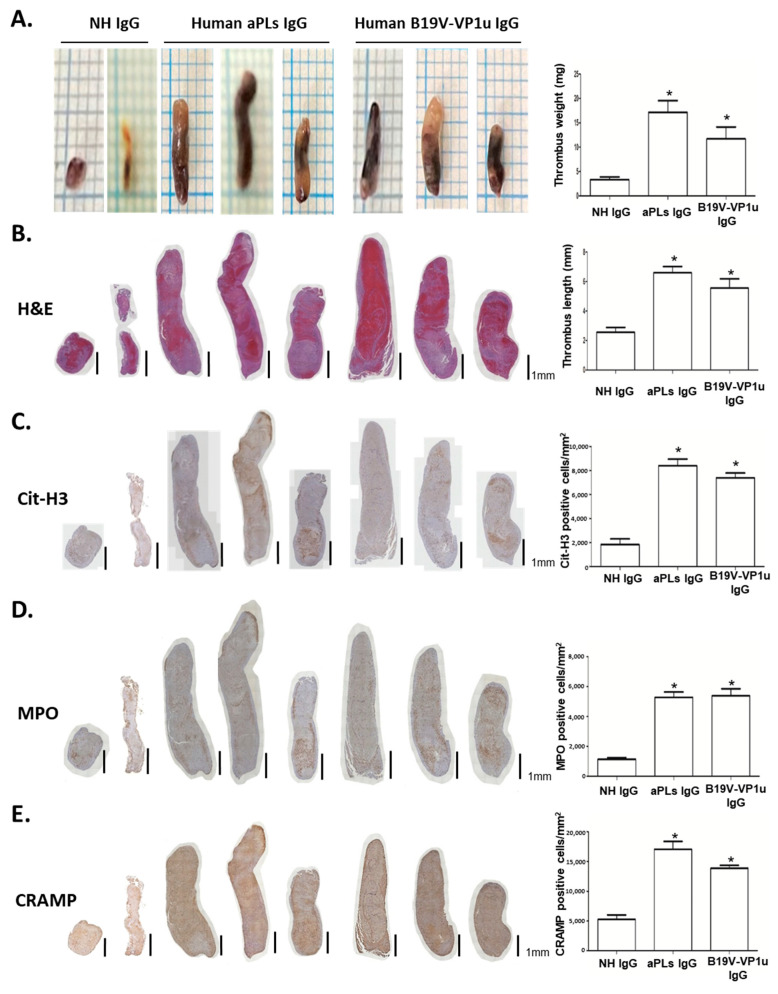
Human aPLs and anti-B19V-VP1u IgG promote venous thrombosis in C57BL/6 mice with inferior vena cava ligation. (**A**) The representative images of thrombus from the mice treated with normal human IgG (NH IgG), human aPLs IgG, and human anti-B19V-VP1u IgG. Sections of the thrombus stained with (**B**) H&E, (**C**) anti-citrullinated histone H3 (Cit-H3), (**D**) MPO, and (**E**) CRAMP. Scale bar = 1 mm. The right panel showed the quantified results of thrombus weight, length, and positive cells of Cit-H3, MPO, and CRAMP signal. The symbol * indicate a significance (*p* < 0.05) as compared to the NH IgG. *p*-value is calculated by one-way ANOVA followed by Tukey’s multiple comparison test.

## Data Availability

The original contributions presented in this study are included in this article; further inquiries can be directed to the corresponding author.

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
