# Peer review of "Parvovirus B19 Infection Is Associated with the Formation of Neutrophil Extracellular Traps and Thrombosis: A Possible Linkage of the VP1 Unique Region"

_ijms, 2024, doi:10.3390/ijms25189917_

Round 1

Reviewer 1 Report

Comments and Suggestions for Authors

The manuscript by Tzang et al demonstrate the role of parvovirus B19-induced antibodies, specifically VP1 unique region, in the induction of NETs and its influence in thrombosis. Overall, the work has interesting findings that could be relevant to prevent post-infection consequences. Nevertheless, some points were raised:

Major points:

1) the authors demonstrate that ROS is induced by B19V-VP1u IgG, possibly contributing to NETosis. However, to claim this statement, blockers of ROS must be used.

2) the authors demonstrate the formation of thrombi after B19V-VP1u IgG administration in vivo. As they demonstrated that 8-Br-cAMP and CGS216800 were able to block NETosis in vitro, I wonder if they could also be beneficial to prevent the thrombosis in vivo.

3) a well-stablished technique to detect NETs release is measuring MPO-DNA complex by ELISA. Using this technique to measure NETs in vitro as well as in vivo would support the B19V-VP1u IgG-induced NETosis concept. 

Minor points:

1) Hoechst signal in the middle panel of figures 1 and 2 is too strong, not allowing a good visualization of the cells.

2) The figures are missing the statistical analysis description.

3) Improvement of the thrombi photos quality in figures 6 and 7 is needed.

Reviewer 2 Report

Comments and Suggestions for Authors

I congratulate the authors to the very small and more or less identical variability in nearly any parameter. Especially, in NETs formation which is in our hands and in the hands of other researchers so variable. Of note, in some graphs there is no visible variability... 3 independent experiments with totally the same results... interesting.

Most of us, however, calculate the % of neutrophils that produce NETs instead of some SYTOX Green signal.

Why are the authors applying different NETs production inhibitors? This should be explained in the Introduction and interpreted in the Discussion.

The authors have to use real neutrophils. Either from human volunteers or from mice. The HL60 cells are not ideal.

The fact that IgG decreases the thrombus size means that these antibodies can be used for the treatment?

Why is the model needed at all? Do the antibodies not stimulate thrombosis on their own?

Why are the Methods between the Discussion and the Conclusion?

How can one "manage or address" the viral infection?

Round 2

Reviewer 1 Report

Comments and Suggestions for Authors

The authors provided discussions about some of the experiments requested. However, this reader believes that at least the in vivo treatment with 8-Br-cAMP or CGS216800 is necessary to make the claim that they are beneficial to prevent thrombosis in the context of B19V-VP1u IgG-induced NETosis.

Reviewer 2 Report

Comments and Suggestions for Authors

The issues I identified were not solved by the revision. Some cannot be.

Round 3

Reviewer 1 Report

Comments and Suggestions for Authors

The authors claimed to have no resources to perform the experiment. Nevertheless, they added comments into the manuscript that highlight the need to demonstrate the mechanism in vivo. As their findings is still novel and relevant, I believe that the manuscript can be accepted.
